# Beauty versus Health—How Eyelash Extensions May Affect Dry Eye Disease?

**DOI:** 10.3390/jcm13113101

**Published:** 2024-05-25

**Authors:** Christina N. Grupcheva, Dimitar I. Grupchev, Natalya Usheva, Lora O. Grupcheva

**Affiliations:** 1Specialized Eye Hospital, Medical University of Varna, 55 Marin Drinov Str., 9010 Varna, Bulgaria; dgrupchev@gmail.com (D.I.G.); loragrupcheva@gmail.com (L.O.G.); 2Faculty of Public Health, Medical University of Varna, 9002 Varna, Bulgaria; nataly_usheva@mu-varna.bg

**Keywords:** eyelash extensions, dry eye, artificial lashes, ocular surface, cosmetic procedures

## Abstract

**Background:** Eyelash extensions (EEs) are among the most popular cosmetic procedures today. There is no prospective study demonstrating how this procedure affects the ocular surface and eye dryness in particular. The goal of this study is to evaluate the effect of EEs removal on dry eye symptoms and signs. **Materials and Methods:** The subjects were prospectively recruited from routine clinical examinations for dry eye complaints. Only subjects with an OSDI score above 31 were included in the study. The subjects also planned to have the EEs removed and agreed to abstain from makeup use and new cosmetic procedures for 4 weeks. The presence of dry eye was evaluated by an OSDI questionnaire, and objectively by tear breakup time (TBUT), staining (Oxford scale) and blinking intervals. All tests for dry eye were performed at baseline and 4 weeks after EEs removal. **Results:** The mean age of our patients, all female, was 28 years. The size and type of EEs was diverse. The decision process was mainly based on appearance and models. None of the subjects had any health conditions. The mean result from the score from the OSDI questionnaire at the baseline was 33.4 and improved to 26.7 points 4 weeks after EEs removal. Objectively, the mean TBUT increased from 11.25 to 13.96 s. For the same period, the blinks increased by two per minute, and the staining was reduced by 1.0 grade. **Conclusions:** Removal of EEs improves the symptoms and the objective signs of dry eye. The most popular beauty procedure regarding eyelashes might not be innocuous to eye health.

## 1. Introduction

Individual artificial eyelashes, known as eyelash extensions (EEs), are a contemporary trend growing in the beauty industry. The users believe that EEs enhance natural beauty with less effort, which has led to an increase in the popularity of this semi-permanent cosmetic option among women, especially the younger age groups. The procedure includes the gluing of individual artificial, silk or mink fibers to each natural eyelash and, depending on the growth cycle, EEs may last 4–8 weeks [1]. According to the latest TFOS lifestyle report, Impact of Cosmetics on the Ocular Surface, artificial eyelashes have the potential to compromise the ocular surface [2]. The effects of EEs associated with dry eye might include, but are not limited to, reduced blinking, incomplete blinking, incomplete closure during sleep, blockage of the glands, retention of dust, allergic reactions, mechanical effects and others [3]. Therefore, artificial eyelashes might play a role in the development of dry eye symptoms over time or be a reason for exacerbation of an already present disease [4,5,6]. Moreover, very often EEs are the reason for neglected eye hygiene and abstention from proper lid cleaning. The EE specialists’ advice is to protect the extensions from water when showering, avoid washing the eyes with warm water and abstain from any lid hygiene procedures, except the ones specifically designed for EEs. It is important to note that the eyelashes have a dense array of sensory terminals, which play an important role in the blink reflex and corneal protection and might be also affected by EEs [7]. All these facts are more of an empirical clinical observation than properly analyzed research findings. In fact, a peer literature search finds only 18 references containing the keywords “eyelash extensions”. They can be grouped as follows:-local complications related to EEs, including keratitis [5,8];-local and systemic allergic reactions related to the glue [3,9,10,11,12,13,14,15,16];-epidemiology and habitual tendencies [1,6,17,18,19];-miscellaneous [20,21].

There are some studies in the USA, Africa (Nigeria and Ghana) and Asia (Japan and China), which demonstrate the side effects and significant complications related to EEs [1,18,19]. A recent report on EEs used by female students in Nigeria highlighted 73.3% complications, including itching (45.8%), redness (45.5%), pain (43.9%) and heavy eyelids (41.6%) [17]. Interestingly, there is another study with a conclusion that the effects of EEs are only temporary [11]. This limited information was the main drive to look in a controlled manner to correlate dry eye findings with EE usage. As there was a prior attempt to look for signs of ocular surface damage 1 h, 1 week and 1 month after EEs, we decided to evaluate the benefits after EEs were removed [18].

We designed a study with the purpose of analyzing the EE effect on ocular dryness utilizing an OSDI questionnaire, tear breakup time (TBUT), and corneal and conjunctival staining as outcome measures.

## 2. Materials and Methods

The subjects were prospectively recruited from routine clinical examinations for dry eye complaints. Only subjects with an OSDI above 31 were included in the study. In addition, they had to comply with the following criteria: uneventful artificial eyelash usage for more than 8 weeks, no recent (within 2 weeks) touch-up procedures, no additional cosmetic procedures, no additional makeup use, no contact lens use, abstention from artificial tear substitutes 8 weeks prior and for the time of the study, no additional ocular or systemic diseases. All subjects had most of their EEs in place and with proper position, and patients with misdirected lashes or mechanical milphosis were excluded from the study (Figure 1A,B). The subjects also planned to have the EEs removed and agreed to abstain from makeup use and new cosmetic procedures for 4 weeks.

From the 120 screened EE users visiting the Specialized Eye Hospital with dry eye complaints but no serious complications, only 34 subjects aged 21–38 fulfilled the inclusion criteria. Subsequently, 31 completed the study, as 3 of the subjects decided that they needed new EEs during the 4-week washout period. The subjects filled out the OSDI questionnaire, followed by comprehensive eye examination, including fluorescein TBUT in a standardized manner, mean number of blinks per minute, and corneal and conjunctival staining. The study and procedures followed the Declaration of Helsinki and were approved by a local ethics committee, with all subjects signing an informed consent form. The EEs were removed by an application specialist selected by the subject, and the same tests were repeated 26–28 days after the removal procedure. During this washout period, the subjects abstained from any periocular procedures, including makeup, and any professional or home cosmetic procedures around the eyes.

The rationale for EE usage was evaluated by a Knowledge and Consideration Questionnaire exploring the information acquired about individual parameters, such as EE characteristics (material, glue, thickness and length); the decision process based on looks, price, health and cautiousness; and the EE maintenance/touch-up intervals followed by each user. The options for material were the well-known ones: mink, silk, synthetic and no information/knowledge. There were 3 options in terms of length: 12 mm (the accepted standard), as long as possible and no information. Three options were also available in regard to thickness: own lashes, thicker and no information. As the number of eyelashes is crucial for the health consequences, the options given to the subjects included more detail as follows: 1:1, 1:2, 1:3, as many as possible and no knowledge. As drives for the individual’s decision, looks and appearance, price and health were given as options. Last but not least, the subjects were supposed to define their maintenance intervals as follows: less than 2 weeks, 2–3 weeks and longer than 3 weeks.

In order to evaluate dry eye, an OSDI questionnaire was used, utilizing a validated Bulgarian translation. Only objective parameters from one eye per subject were taken. For the purpose of the study, the dominant eye was used, and this was determined by “fogging” with a +1.0 Dpt lens. The number of blinks was counted by a side observer when the patient was interviewed by the doctor. The TBUT was performed with fluorescein strips (1 mg BioGlo^TM^, HUB Pharmaceuticals, Farmington Hills, MI, USA), gently wetted with sodium chloride and applied in the lateral part of the lower fornix. After 30 s, the TBUT was performed 3 times, and the average was taken as a result. The staining was evaluated according to the Oxford scale. All tests for dry eye were performed at baseline and 4 weeks after the EEs were removed. The clinician also observed and noted the eyelash health, including milphosis, misdirections, retention of dust and Demodex infestations. The term milphosis was used because madarosis is reserved for eyelash and/or eyebrow loss, and for our patients, the brows were mostly treated by microblading.

The collected data are presented using descriptive statistics for quantitative data–mean, standard deviation and standard error of the mean. Statistical analysis includes comparison of the means of the three indicators (before and after) using the paired samples *t*-test method.

## 3. Results

The mean age of our patients, all female, was 28 years. The size and type of the EEs were diverse, and most of the subjects did not have any knowledge about the material of EEs nor the glue used. The majority of the tested young women preferred more volume, which meant two, three or even more artificial lashes, to one natural lash. The decision process was mainly based on appearance and models (Table 1). None of the subjects had any health considerations. The cost was acceptable and not shown to be among the main drives. Objectively, because of no recent touch-up procedures (a minimum of 2 weeks), the EEs did not look very impressive (Figure 1). Many of the subjects presented with misdirection, local milphosis, retention of debris and cosmetics, and even significant Demodex infestation of their own lashes. When the eyelashes were removed, most of the subjects were not satisfied with their look, but significant improvement was encountered 4 weeks after. The mean result from the OSDI questionnaire (baseline 33.4) decreased to 26.7 points. There were 27 right eyes and 4 left eyes, as the majority of the patients had strong right eye dominance. Objectively, the mean TBUT increased from 11.25 to 13.96 s. For the same period, the blinks increased by two per minute, and the staining was reduced by 1.0 grade. All objective outcomes correlated with the subjective assessment from the OSDI questionnaire. The results are presented in Table 2. The statistical analysis demonstrated the significance of eyelash removal on all outcome parameters. Further correlation analysis showed no significance of age on the outcome parameters, except reduced blinking.

## 4. Discussion

The modern world is not imaginable without artificial EEs. There is a wider interest by diverse societal groups, and therefore, a range of application specialists (from highly professional makeup artists to amateurs) are practicing this procedure [1]. Although there are no real data for the frequency and sustainability of this semi-permanent cosmetic procedure, in everyday life we see more and more female subjects utilizing this cosmetic option, not only in the public space but also in ophthalmic offices. Very often, those subjects visit eyecare practices with complaints and/or complications [17]. More than 10 years ago, Amano et al. tried to raise scientific and clinical awareness that artificial eyelashes are harmful to the ocular surface, mainly because of the glue used, but also because of traumatic events and allergic reactions [3]. Over a decade later, there are no more than 30 publications (including published abstracts) evaluating the negative effects of EEs on the ocular surface. The primary argument of eyecare practitioners against EEs is the detrimental effect on anatomy, mainly concerning the destruction of one’s own lashes and reduced lid hygiene [2,6,11,17]. The natural life cycle of eyelashes involves three phases: anagen, catagen and telogen, but with a long resting telogen phase. Therefore, eyelashes grow only to about 10 mm and then shed naturally (2–4 per day) [22]. This explains the reason for fortnightly to monthly maintenance of EEs. Although the mentioned considerations are reasonable and logical, potential clients concentrate on the benefits related to looks. 

Our study demonstrated that the users had very limited knowledge about the material of the extensions and the glue used for adhesion. The main interests are concentrated on looks, easy maintenance and, for a small number of subjects, also on the price. We conducted research in the local studios, and all glues used there contained cyanoacrylate (glue), polymethyl-methacrylate (additional adhesive), hydroquinone (phenol derivative to prevent fast drying) and carbon black (to keep the color). The users did not even consider the possible health consequences; they just wanted to have a “superstar look” and had no knowledge that the aforementioned components might cause toxic and allergic reactions and ocular surface damage. In the limited literature publications, there is evidence that the glue might be the principal factor for ocular surface problems, such as keratoconjunctivitis, blepharitis and even erosions [3,12,13,14,15]. According to the published research, the glue also frequently contains formaldehyde, which is proven to cause allergic and toxic reactions of the conjunctiva and periocular skin [3,14]. 

The outcome measures of the current study were based on an OSDI questionnaire, classic fluorescein TBUT, staining and frequency of blinking. Of course, the study has a number of limitations, including more sophisticated ocular surface analysis with contemporary technology and long-term follow-up. However, all of the followed parameters demonstrated significant improvement 4 weeks after EEs removal. The subjects used diverse EEs with different length, material and number of lashes applied per natural lash, but were controlled for extra cosmetics and contact lenses. Therefore, it is reasonable to conclude that the observed positive outcomes were mainly related to EEs removal. Importantly, in four weeks, the staining of the cornea was significantly less. This is a very important finding, because EEs are associated with inflammation and keratitis [5,8]. The epithelial barrier might be broken by the mechanical effects, as well as by the chemical glue or removal solvents [2]. Further experimental studies are required to evaluate the effects of all these components on ocular surface homeostasis. 

Another interesting observation of our study is the interest of more than two-thirds of consumers to have EEs with more volume. This means that to one eyelash usually 2+ extensions are glued. In the most impressive published review on eyelash anatomy and pathology, Aumond and Bitton highlighted the importance of structure, volume and length of the eyelashes for ocular surface health [22]. In this comprehensive paper, the authors paid special attention to the aerodynamics of the eyelashes and their role for ocular surface protection [22]. According to an experimental study by Amador et al., the optimal length of the eyelash should be 1/3 of the width of the palpebral fissure, or a mean of 12 mm. In our study, less than 20% of the subjects were aiming for this length. The majority of our participants had strong preferences for longer lashes or had no knowledge about the exact parameters. However, the experiments of Amador et al. demonstrated that longer lashes are associated with more evaporation; therefore, dry eye risks are higher [23]. 

Furthermore, in our tested group of users, synthetic fibers were most frequently used, which also increases the weight of each extension. This is probably the most rational explanation regarding milphosis, demonstrated in Figure 1, which in fact is the main drive for continuation of the procedure. A study by Kadri et al. looked at the effect of mascara on eyelash milphosis [24]. The authors proved that everyday application of mascara, as well as cleaning waterproof mascara with water and water-based products, leads to statistically significant milphosis [24]. Hypothetically, these effects might be also attributed to EEs, as they are a semi-permanent, long-term procedure with hydrophobic components. Although we did not use complications as outcome measures, during the study we noted negative effects such as milphosis, misdirections of the EE, Demodex infestation and a lot of uncleaned debris in the roots of the subjects’ own eyelashes. Previously, other authors have highlighted significant acute complications among EE users [17]. In a comprehensive literature review, Masud et al. found in the published literature 42 cases of allergic blepharitis, 4 of keratoconjunctivitis, 3 of conjunctival erosion, 2 of contact dermatitis, 1 of bacterial keratitis and 1 of subconjunctival erosion in 2019 [11]. A recent study from Benin City, Nigeria, has demonstrated that side effects after application of EEs increase from 16% (before EEs application) to 54% (after EEs application) [17]. Regardless of those published observations and clinical practice experiences, there is still no prospective, controlled study in the literature.

Eyecare practitioners must be aware of the potential risks associated with artificial eyelashes. Previously published research mentions that public awareness regarding the health consequences of EEs is also urgently needed [4]. However, as with other hazards in life, the outcome of more expressive eyes overcomes any health consideration. Unpublished observations of the authors show that, even after complications, some users prefer to come back to EEs. There is no published study on the effect of EEs taking into account the size, type and material. The so-called “Russian” volume lashes include longer, denser and heavier EEs. This style definitely has a more detrimental effect on blinking and, therefore, is a significant factor for dry eye development or exacerbation. In the training programs of EE application specialists, an eyelash weight calculator is often available. However, the calculator is calibrated according to the thickness (strength) of natural eyelashes. The weight of EEs is having an effect not only on the lash follicles but also on the mechanics of the upper eyelid, especially in high-volume EEs. Additional research is needed to define the effect of artificial eyelashes on various aspects of ocular health and the long-term effect on lid movements.

A recent TFOS workshop was dedicated to environmental challenges [2]. Cosmetics and cosmetic procedures were recognized as such a challenge with a significant effect on the ocular surface. More than 50 cosmetic products and procedures were recognized [2]. The user is mostly of the female gender, although for some, gender does not matter. Having EEs applied is, however, a procedure exclusively used by women and especially urbanites and such at a younger age. This possesses a very high potential for the amplification of factors provoking dry eye. The current study just demonstrated that removal of EEs improves the OSDI index and TBUT, and decreases the staining of the ocular surface. The assumption is that application of EEs leads to the opposite effect. In fact, a recent, elegant study by Han et al. has demonstrated a similar effect but in prospect of a follow-up after EEs application. The authors observed the dynamics of OSDI, conjunctival vascular density, tear meniscus height (TMH), noninvasive TBUT, bulbar redness, meibography, lipid layer thickness and corneal staining only for the right eyes [18]. All tests were performed 1 h, 1 day, 1 week and 1 month after EEs application. The authors found that the feeling of having a foreign body and the staining were highest an hour after the application of eyelashes. Tear breakup time and TMH were lowest at the 1-week visit. The rest of the parameters did not demonstrate a significant change. Therefore, the authors concluded that EEs disturb ocular surface homeostasis, but perhaps the effect is strongest during the first week [18].

Another very interesting subject is the combination of contact lenses with EEs. Most of the sites advise that contact lenses be removed before the procedure and then inserted 1 h to 1 day after the application of EEs. Interestingly, in the peer-reviewed literature, there is no published study on the effect of EEs on contact lens discomfort. However, there is a significant number of publications dedicated to dry eye and contact lenses. Taking into consideration the effect of EEs on tear film and meibomian glands, one may speculate that the symptoms of dry eye might be exacerbated by the application of EEs. Even more, the glue might have chemical or mechanical effects on certain contact lens materials. There is an interesting study demonstrating the effect of mascara on ocular surface and meibomian gland damage [25]. In the same study, the authors hypothesized that meibomian gland loss in makeup users might be a result of pigment retention and migration in delicate ocular tissues [25]. 

Beauty and the beauty industry are a significant part of our social life. Eyes have long been an established symbol of beauty, and there are some studies analyzing the social preferences in terms of eyelash length. Pazhoohi et al. performed an interesting study using combinations of eyelash length and ethnicity and discovered that the preferences demonstrate a U-shaped distribution, namely, very short and very long eyelashes are not liked by the majority of tested individuals [26]. Interestingly, they also found that ethnicity also matters, and for Europeans, average (not longer) eyelash lengths are better accepted [26]. Regardless of the social appreciation of beauty, a healthy lifestyle and eye health should be a human priority. The balance must be found by eyecare practitioners, consumers and other service providers. Based on the published literature, there has also been a long-established interest from dermatologists in the effect of eye cosmetics and especially EEs on periocular skin [27]. Somehow, the cosmetic application around the eyes is a watershed area, and the safest approach is an interdisciplinary involvement of eyecare practitioners, dermatologists, and medical training of beauty specialists. Recently, in our practice, there has been an increasing upward trend of eye complications from cosmetic procedures, several of which are already legal cases with court decisions against the application specialist. There is even a paper titled “What Lash Stylists and Dermatologist Should Know?”, but there should also be improvement in the knowledge of the eyecare practitioners who are managing ocular complaints, problems and complications [10]. The scientific questions to be answered in the future regarding the EE include, but are not limited to:-How would long-term ocular surface homeostasis and dry eye in particular be affected by EEs?-What is the short- and long-term effect of EEs on the eyelash cycles and health?-How may a combined use of different types of contact lenses with EEs affect the ocular surface?-What is the effect of EEs on the mechanics and position of the upper lid?-Which are the most common complications of EEs, and how can they be prevented?

## 5. Conclusions

Little is known in the literature about the effects of EEs on the ocular surface and its homeostasis. This current study demonstrates that the removal of EEs improves the symptoms and objective signs of dry eye. Today, occasional or semi-permanent EE application is the most popular beauty procedure regarding eyelashes, but it seems to be significantly associated with complaints and eye health issues. Further research is needed to demonstrate the exact effects of diverse EEs on the ocular surface. Collaborative research led by eyecare practitioners should involve a dermatologist and an eyelash application specialist/stylist. Last but not least, improving the public awareness is a very important task, but solid research is needed to draw conclusions and even regulations regarding EE procedures.

## Figures and Tables

**Figure 1 jcm-13-03101-f001:**
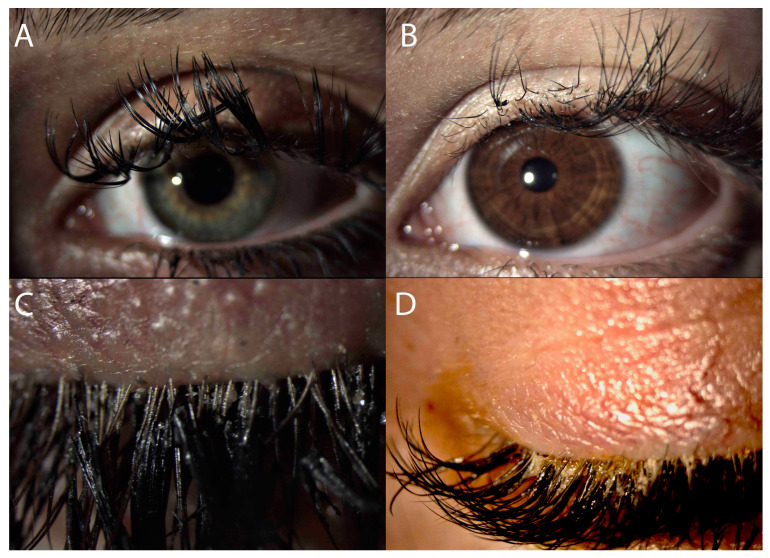
Misdirected eyelashes (**A**), mechanical milphosis (**B**), Demodex infestation in the roots of own lashes (**C**) and uncleaned material on the top of the eyelashes (**D**).

**Table 1 jcm-13-03101-t001:** Results from the Knowledge and Consideration Questionnaire exploring the information acquired about individual parameters, such as EE characteristics (material, glue, thickness and length); the decision process based on looks, price, health cautiousness, etc.; and the EE maintenance/touch-up intervals followed by the individual user. The answers of all 31 subjects are included.

Characteristics per All Tested Subjects (N = 31)	Specified Features of Each Individual Characteristic
Material	Mink (N = 2), Silk (N = 3), Synthetic (N = 2), Does not know (N = 24)
Length	12 mm or less (N = 6), As long as possible (N = 7), Does not know (N = 18)
Thickness	Thickness of own lashes (N = 6), Thicker than own lashes (N = 3), Does not know (N = 22)
Number of fibers per natural eyelash	1:1 (N = 6), 1:2 (N = 7), 1:3 (N = 8), As many as possible (N = 4), Does not know (N = 6)
Drive for decision	Looks and appearance (N = 29), Price (N = 2), Health (N = 0)
Maintenance intervals	Less than 2 weeks (N = 2), 2–3 weeks (N = 28), 3+ weeks (N = 1)

**Table 2 jcm-13-03101-t002:** Average results from the OSDI questionnaire and clinical tests (blinking frequency, TBUT and staining) performed at baseline and 4 weeks after, for 31 subjects included in the study. Only 1 eye per patient, the dominant eye, was evaluated objectively (27 right and 4 left eyes).

Parameter	Baseline	In 4 Weeks	Statistical Significance
OSDI	33.4	26.8	<0.0001
Blinks per minute	6.7	9.8	<0.0001
TBUT	11.3	14	<0.0001
Staining	2.5	1.5	<0.0001

## Data Availability

Data are contained within the article.

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
