# Peer review of "Beauty versus Health—How Eyelash Extensions May Affect Dry Eye Disease?"

_jcm, 2024, doi:10.3390/jcm13113101_

Round 1
Reviewer 1 Report
Comments and Suggestions for Authors
Prior to any consideration for publication, this study should address the following: Comments of major importance: -The study does not include any power analysis that could determine if the sample size is adequate. Comments of minor importance:
Please rephrase/replace words in the following lines:
Line 18 eye dryness instead of "dry eye"
Line 21 OSDI score instead of just "OSDI"
Line 22 the presence of dry eye disease
Line 26 any health "conditions" instead of "consideration"
Line 27 score from the OSDI
Line 42 are a contemporary trend growing in the beauty industry.
Line 50 dry eye disease/symptoms
Line 98 please use the word "used" instead of "applied"
Line 156 needs to be rephrased
Line 202 "debris" instead of "dirt"
line 203 rephrase the word "elegant"
Line 270 please replace "this current study" instead of "a current study"
Furthermore, Lines 237-245 are not adding any knew insight, Lines 247-260 are not adding any knew insight
There is also a typographical error in line 41
the font is not the same across the entire paper
Discussion: The limitations of this study should be mentioned
Paragraph 6.Patients does not need to be included
In general, the manuscript is interesting, timely and the ideas flow logically.
Comments on the Quality of English LanguageOverall, the quality of English language is adequate, however some words would be preferable to be rephrased as mentioned in the comments and suggestions for the authors
Author Response
Dear reviewer,
We gratefully corrected the manuscript according to your remarks.
The answer of our statistician, regarding your principle question is: "Calculating the required sample size at 80% test power establishes the need to include a minimum of 25 participants. Even after using the bootstrapping technique, the differences remained statistically significant (<0.001)."
Comments of minor importance:
Please rephrase/replace words in the following lines:
Line 18 eye dryness instead of "dry eye" - DONE
Line 21 OSDI score instead of just "OSDI" - DONE
Line 22 the presence of dry eye disease - DONE
Line 26 any health "conditions" instead of "consideration" - DONE
Line 27 score from the OSDI - DONE
Line 42 are a contemporary trend growing in the beauty industry - DONE
Line 50 dry eye symptoms - DONE
Line 98 please use the word "used" instead of "applied" - DONE
Line 156 needs to be rephrased : "There is a wider interest by diverse societal groups and therefore a range of application specialists (from highly professional makeup artists to the amateurs) are practicing this procedure."
Line 202 "debris" instead of "dirt" DONE
line 203 rephrase the word "elegant"- "comprehensive"
Line 270 please replace "this current study" instead of "a current study" DONE
Furthermore, Lines 237-245 are not adding any knew insight, Lines 247-260 are not adding any knew insight
There is also a typographical error in line 41 DONE
the font is not the same across the entire paper - arranged
Discussion: The limitations of this study should be mentioned - DONE
Paragraph 6.Patients does not need to be included
In general, the manuscript is interesting, timely and the ideas flow logically.
Thank you very much for the detailed comments. We addressed them and this definitely improved the quality of our manuscript. Let express our gratitude for your time and efforts! We really appreciate it! Kindest regards on behalf of all authors!
Reviewer 2 Report
Comments and Suggestions for Authors
This study is topical and very interesting. There is a growing interest in the effects of cosmetic treatments on the ocular surface. This study investigated the effects of removal of eyelash extensions on the ocular surface and found that the removal of eyelash extensions improved the signs and symptoms of the ocular surface. While the results seem intriguing, the following comments need addressing:
1. Line 17 “There is no clear evidence how this procedure affects the ocular surface and dry eye in particular”.
There have been previous articles conducted on the study of undergo eyelash extensions on the ocular surface, which is also elaborated in the introduction. So the expression in this sentence is not correct.
2. Line 30 “Removal of EE improves the symptoms and the objective signs of dry eye. The most popular beauty procedure regarding eyelashes might not be innocuous to eye health”.
This study found that removal of EE improves symptoms and objective signs, why later said its not harmful to health.
3. Line 65-67 “As 65 there was a prior attempt to look for markers of ocular surface damage 1 hour, 1 week, and 1 month after EE, we decided to evaluate the benefits after EE were removed”.
The article cited was not looking for markers, but rather exploring the effects of eyelash extensions on the ocular surface.
Han J, Xie Z, Zhu X, Ruan W, Lin M, Xu Z, et al. The effects of eyelash extensions on the ocular surface. Cont Lens 342 Anterior Eye. 2024:102109.
4. Line 41 “tttttttct1. Introduction” wrongly written.
5. Line 73 Does this study standardise the number of previous eyelash extensions?
6. Line 83-85 “The EE were removed by an application specialist selected by the subject and the same tests were repeated 26–28 days after the removal procedure”.
This study examines whether the time that subjects have had their eyelash extensions before removing the false eyelashes is uniform. If the extensions had been in place for more than 4 weeks, the false eyelashes were almost completely removed and the study was more likely to have negative results. Also, the previous statement "no recent (within 2 weeks) touch-up procedures" suggests that the subjects' extensions had been in place for a longer period of time and that the false eyelashes may have been lost.
7. What are the statistical methods used in this study?
8. Line 103-104 “All tests for dry 103 eye were performed at baseline and 4 weeks after the EE were remove”.
Why did you choose to assess at 4 weeks and only baseline and 4 week assessments? Was there any impact on the ocular surface as a result of this operation of eyelash extension removal? How was the removal performed and what reagents were used for the removal? Why was an examination not performed after the removal?
9. table 2 fonts not standardized.
10. Are the parameters in Table 2 expressed as averages? Suggest to make it clear.
11. Line 230 font is not standardized.
Comments on the Quality of English LanguageModerate editing of English language required
Author Response
We thank the reviewer for the most useful suggestions. Our amendments are highlighted below:
- Line 17 “There is no clear evidence how this procedure affects the ocular surface and dry eye in particular”. Changed: "There is no prospective study demonstrating how this procedure affects the ocular surface and eye dryness in particular."
There have been previous articles conducted on the study of undergo eyelash extensions on the ocular surface, which is also elaborated in the introduction. So the expression in this sentence is not correct.
- Line 30 “Removal of EE improves the symptoms and the objective signs of dry eye. The most popular beauty procedure regarding eyelashes might not be innocuous to eye health”.
This study found that removal of EE improves symptoms and objective signs, why later said its not harmful to health. "might not be innocuous" - mean might be harmful
- Line 65-67 “As 65 there was a prior attempt to look for markers of ocular surface damage 1 hour, 1 week, and 1 month after EE, we decided to evaluate the benefits after EE were removed”.
The article cited was not looking for markers, but rather exploring the effects of eyelash extensions on the ocular surface. "markers" was exchanged with "signs"
Han J, Xie Z, Zhu X, Ruan W, Lin M, Xu Z, et al. The effects of eyelash extensions on the ocular surface. Cont Lens 342 Anterior Eye. 2024:102109.
- Line 41 “tttttttct1. Introduction” wrongly written. Omitted
- Line 73 Does this study standardise the number of previous eyelash extensions? NO it is not possible as all application specialist have their own style.
- Line 83-85 “The EE were removed by an application specialist selected by the subject and the same tests were repeated 26–28 days after the removal procedure”.
This study examines whether the time that subjects have had their eyelash extensions before removing the false eyelashes is uniform. If the extensions had been in place for more than 4 weeks, the false eyelashes were almost completely removed and the study was more likely to have negative results. Also, the previous statement "no recent (within 2 weeks) touch-up procedures" suggests that the subjects' extensions had been in place for a longer period of time and that the false eyelashes may have been lost.
To clarify the issue we added missing part of our inclusion criteria: All subjects had most of their EE in place and with proper position and patients with misdirected lashes or mechanical mylphosis were excluded from the study (figure 1 A and 1 B).
- What are the statistical methods used in this study?
The collected data are presented using descriptive statistics for quantitative data - mean, standard deviation and standard error of the mean. Statistical analysis includes comparison of means of the three indicators (before and after) using Paired samples t-test method.
- Line 103-104 “All tests for dry 103 eye were performed at baseline and 4 weeks after the EE were remove”.
Why did you choose to assess at 4 weeks and only baseline and 4 week assessments? Was there any impact on the ocular surface as a result of this operation of eyelash extension removal? How was the removal performed and what reagents were used for the removal? Why was an examination not performed after the removal?
The group is very sensitive to medical interventions, and therefore we had to be careful with number of evaluations in order to keep the participants. On the basis of unpublished research the day after removal is mostly associated with significant OS staining and TBUT is almost not possible.
- table 2 fonts not standardized. Done
- Are the parameters in Table 2 expressed as averages? Suggest to make it clear. - Done thank you
- Line 230 font is not standardized. Done
Thank you very much for the valuable comments!
Kindest regards on behalf of all authors!
Reviewer 3 Report
Comments and Suggestions for Authors
As per attached pdf file

Author Response
Dear Reviewer,
We are most grateful for your comments and implemented most of your suggestions in the introduction and discussion. As we had no knowledge(neither the patient did) about the exact composition of glue and the remover we can not address this issue, but certainly we are planning on additional studies. This is also valid for OSA application as we just purchased one and hope to have some more detailed data soon.
Most grateful we are for the reference of Halata et al, as it provides extremely important information about the neurosensory role of eyelashes.
One more time deep thank you for the constructive critiques, which definitely improves the quality of our manuscript.
On behalf of all authors